# The indirect effect of mRNA-based COVID-19 vaccination on healthcare workers' unvaccinated household members

Jussipekka Salo [1,2,3,4], Milla Hägg [1,2,3,4], Mika Kortelainen [2,4,5,6], Tuija Leino[7], Tanja Saxell[2,4], Markku Siikanen [2,4] & Lauri Sääksvuori [8,9,10 ✉]

Mass vaccination is effective in reducing SARS-CoV-2 infections among vaccinated individuals. However, it remains unclear how effectively COVID-19 vaccines prevent people from spreading the virus to their close contacts. Using nationwide administrative datasets on SARS-CoV-2 infections, vaccination records, demographics, and unique household IDs, we conducted an observational cohort study to estimate the direct and indirect effectiveness of mRNA-based COVID-19 vaccines in reducing infections among vaccinated healthcare workers and their unvaccinated household members. Our estimates for adults imply indirect effectiveness of 39.1% (95% CI: −7.1% to 65.3%) two weeks and 39.0% (95% CI: 18.9% to 54.0%) eight weeks after the second dose. We find that the indirect effect of mRNA-based COVID-19 vaccines within households is smaller for unvaccinated children than for adults and statistically insignificant. Here, we show that mRNA-based COVID-19 vaccines are associated with a reduction in SARS-CoV-2 infections not only among vaccinated individuals but also among unvaccinated adult household members in a real-world setting.

[1] Department of Economics, University of Helsinki, Arkadiankatu 7 P.O. Box 17, 00014 Helsinki, Finland. [2] VATT Institute for Economic Research, Arkadiankatu 7 P.O. Box 1279, 00101 Helsinki, Finland. [3] Department of Economics, Aalto University School of Business, Aalto University, P.O. Box 21210, 00076 Espoo, Finland. [4] Helsinki GSE, Economicum, Arkadiankatu 7, 00100 Helsinki, Finland. [5] Department of Economics, Turku School of Economics, Rehtorinpellonkatu 3, 20500 Turku, Finland. [6] InFLAMES Research Flagship Center, University of Turku, Turku, Finland. [7] Infectious Disease Control and Vaccinations Unit, Finnish Institute for Health and Welfare, P.O. Box 30, 00271 Helsinki, Finland. [8] Health Science Unit, Tampere University, Arvo Ylpön katu 34, 33520 Tampere, Finland. [9] INVEST Research Flagship Center, University of Turku, Assistentinkatu 7, 20014 Turku, Finland. [10] Finnish Institute for Health and Welfare, Centre for Health and Social Economic, P.O. Box 30, 00271 Helsinki, Finland. ✉email: lauri.saaksvuori@thl.fi

Governments around the world are hoping to lift COVID-19 restrictions as vaccination coverage increases. However, with the limited supply of vaccines and the emergence of new virus variants, decision makers in public health are facing a constant need to update their vaccination and contact restriction strategies. There is currently no consensus about the potential benefits of including all children in mass vaccination programs and much uncertainty about the prospect of achieving herd immunity through vaccination.

COVID-19 vaccines have been shown to be effective in preventing symptomatic and asymptomatic disease among vaccinated adults, both in clinical trials and in nationwide mass vaccination settings[1–4]. Furthermore, there is evidence that vaccinated individuals infected with SARS-CoV-2 have lower viral loads than unvaccinated infected individuals, even though this might have changed after the emergence of the Delta variant[5–8]. However, it remains unclear how effectively COVID-19 vaccines prevent people from becoming infected and spreading the virus to their close contacts, most notably to children and other household members, in real-world circumstances.

This paper provides evidence about the direct and indirect effectiveness of COVID-19 vaccines in reducing infections among vaccinated individuals and their unvaccinated household members. Our analysis exploits the rollout of the mass vaccine program in a large cohort of healthcare workers in Finland, allowing us to estimate the indirect effects of vaccines in a large sample of household members with discordant vaccination status.

We used national databases that record all polymerase chain reaction (PCR)-confirmed SARS-CoV-2 infections and mRNA-based (BNT162b2 by Pfizer-BioNTech or mRNA-1273 by Moderna) vaccine doses administered in Finland up to the beginning of the mass vaccination program among the general working-age population (individuals not belonging to any medically defined risk group or working in the healthcare sector) on April 25, 2021. These data were merged with administrative full-population datasets that include information on each person's occupation and unique identifiers for partners and children living in the same household.

Here, we show that mRNA-based COVID-19 vaccines are associated with a reduction in SARS-CoV-2 infections not only among vaccinated individuals but also among unvaccinated adult household members in a real-world setting. We do not find a decrease in the cumulative incidence of SARS-CoV-2 infections for unvaccinated children in households with at least one vaccinated adult. Taken together, our results suggest that mRNA-based vaccines do not only prevent SARS-CoV-2 infections among vaccinated individuals but also lead to a substantial reduction in infections among unvaccinated adult household members.

## Results

**Study setting and sample**. To estimate the direct and indirect effectiveness of mRNA-based vaccines, we compared the cumulative incidence of SARS-CoV-2 infections between vaccinated and unvaccinated healthcare workers as well as between their unvaccinated partners and children living in the same household (see Supplementary Tables 2 and 3 for the covariate balance between the treatment and control groups). The estimates were adjusted for the state of the epidemic (calendar time), age, sex, occupational group, household size, ethnicity, and geographic area (urban, semi-urban, or rural).

We report vaccine effectiveness estimates by follow-up week after receiving the first and second dose. For each unvaccinated healthcare worker and their family member in the control group, we randomly assigned the beginning of a follow-up period during the observation period. By using this novel approach and controlling for calendar time (week fixed effects), we ensure that the estimates are not biased by the changing nature of the epidemic during the follow-up period. Moreover, to make individuals in the treatment and control groups comparable in terms of their SARS-CoV-2 history, we focused only on persons who had not been infected before their follow-up period. We report vaccine effectiveness estimates from follow-up week 2 onwards because the cumulative incidence of infections is very small in the first 2 follow-up weeks in our sample (Supplementary Fig. 2).

Our sample comprised a total of 265,326 healthcare workers, 128,952 unvaccinated partners of healthcare workers and 169,148 unvaccinated children of healthcare workers. The total number of PCR-confirmed SARS-CoV-2 infections in our sample was 1471 (0.55%), 782 (0.61%), and 820 (0.48%) for healthcare workers, partners, and children, respectively. The mean ages of the healthcare workers, their partners, and children were 44 (SD 14), 45 (SD 12), and 11 (SD 5) years, respectively. Most healthcare workers (86.6%) in our sample were women. A total of 112,496 healthcare workers (42.4%) obtained at least one dose of mRNA-based vaccine during the period from December 27, 2020, to April 25, 2021. The number of double vaccinated healthcare workers during the same time period was 63,986 (24.1% of all healthcare workers).

We observe that the number of infections in different subgroups largely follows the general demographic and occupational profile of our sample. The group of practical nurses is the only occupational group in which the share of infections is somewhat larger than the relative share of the occupational group. Overall, 73.3% of infections are observed among the practical nurses. The most notably overrepresented demographic subgroup in terms of infections relative to the sample share of the subgroup is the group of individuals with a foreign background. While the relative share of individuals with a foreign background is 6.3% in our sample of healthcare workers, their relative share of all infections in our sample is 23.7%. A corresponding overrepresentation of infected foreign individuals is observed among the partners and children of healthcare workers. Supplementary Tables 5–7 report the number of infections in all different subgroups.

**Direct and indirect vaccine effectiveness for adults**. Figure 1 shows our effectiveness estimates for working-age healthcare workers and their unvaccinated partners after the first vaccine dose that was received between the start of the mass vaccination program (December 27, 2020) and April 25, 2021, in Finland. Our estimates imply a gradual increase in direct vaccine effectiveness over time (Fig. 1a): 44.4% (95% CI: 30.4% to 55.6%) 4 weeks and 63.0% (95% CI: 56.3% to 68.7%) 12 weeks after the first dose. We observe that the direct vaccine effectiveness estimates increase over time and stabilize around follow-up week 8. Thus, the stabilization of direct vaccine effectiveness estimates occurs later in our observational study than in several other observational studies[3,4,9,10]. This tendency might be explained by several factors. First, and most importantly, our direct effectiveness estimates (Fig. 1a) include a mixture of individuals who have received either the first or second vaccine dose. The additional protection conferred by the second vaccine dose is expected to be observed, at the earliest, 4 weeks after the beginning of the follow-up as there was a minimum interval of 3 weeks between the first and second dose. Second, the later stabilization of direct vaccine effectiveness estimates might be explained by several contextual differences between the previously reported results and our findings including, among others, different testing strategies, study populations and differences in administrative healthcare register protocols.

Our estimates for unvaccinated partners imply indirect effectiveness of 16.7% (95% CI: –11.9% to 38.0%) 4 weeks and

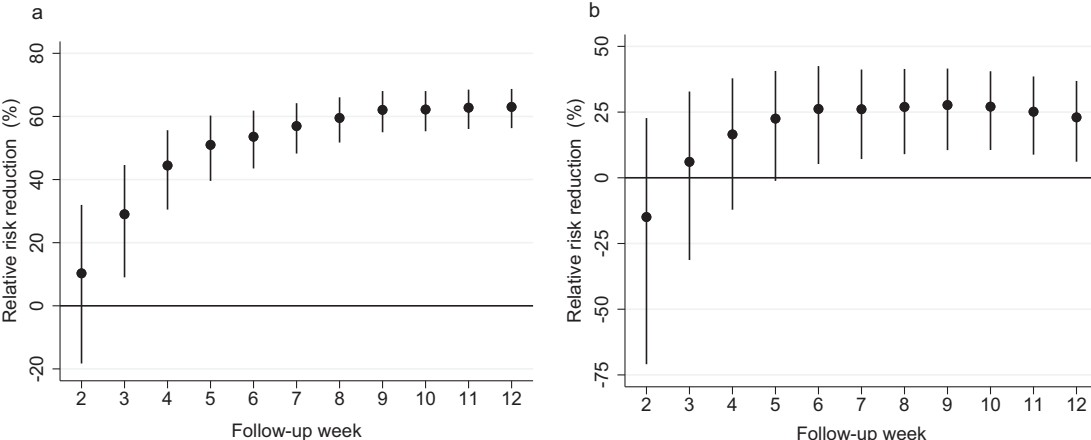

**Fig. 1 Vaccine effectiveness in vaccinated healthcare workers and their unvaccinated partners living in the same household.** This figure plots vaccine effectiveness estimates by week after the first dose of mRNA-based vaccination. The dependent variable is a polymerase chain reaction (PCR)-confirmed SARS-CoV-2 infection as recorded in the Finnish National Infectious Diseases Register. **a** Direct effectiveness estimates (relative risk reduction) for vaccinated individuals compared to the control group, which is constructed by randomly assigning the beginning of a follow-up period for unvaccinated individuals. N = 265,326 healthcare worker observations. **b** Indirect effectiveness estimates (relative risk reduction) for the unvaccinated partners (including cohabiting partners) of vaccinated individuals who lived in the same household as the vaccinated person as of December 31, 2019. N = 128,952 partner observations. The coefficients in both panels are estimated using a log-binomial regression model and individual-date data collapsed to individual-week data. See section Methods, Estimation samples, for details of the sample restrictions and estimation method. The relative risk reduction is presented as a percentage in data points ±95% confidence intervals. The solid black dots show relative risk reduction by week after receiving the first dose of mRNA-based vaccination. The error bars denote the 95% confidence intervals clustered at the individual level using endpoint transformation of the originally estimated confidence intervals.

23.0% (95% CI: 6.2% to 36.9%) 12 weeks after the first dose (Fig. 1b). The indirect effectiveness estimates are substantial and increase gradually, reflecting the corresponding increase in the direct effectiveness estimates. However, as expected, the indirect effects among unvaccinated partners are smaller than the direct effects among vaccinated individuals.

**Indirect effectiveness in children**. Figure 2 shows evidence of the indirect effectiveness of mRNA-based vaccines in preventing SARS-CoV-2 infections among unvaccinated children aged between 3 and 18 years (Fig. 2a) and by age groups: from 3 to 12 years old and from 13 to 18 years old (Fig. 2b). Figure 2 stacks the estimates from follow-up weeks 2–5 into a single average estimate as there are too few PCR-confirmed infections among children (Supplementary Fig. 1) to make reliable inference about the short-term indirect effectiveness of COVID-19 vaccines in this age group. The indirect effectiveness among unvaccinated children aged from 3 to 18 years is estimated to be smaller than among unvaccinated partners (Fig. 2a): −16.3% (95% CI: −65.8% to 18.4%) 2–5 weeks and 6.8% (95% CI: −18.5% to 26.7%) 12 weeks after the first dose.

The indirect effectiveness estimates for 3–12-year-old children are not statistically significantly different from zero (Fig. 2b): −37.1% (95% CI: −127.9% to 17.5%) 2–5 weeks and 8.1% (95% CI: −24.1% to 32.0%) 12 weeks after the first dose. For 13–18-year-old adolescents, the indirect effects are also statistically insignificant and smaller than for unvaccinated partners: 4.8% (95% CI: −47.9% to 38.7%) 2–5 weeks and 4.5% (95% CI: −32.1% to 31.0%) 12 weeks after the first dose. Overall, our results suggest that there is no decrease in the cumulative incidence of SARS-CoV-2 infections for unvaccinated children in households with at least one vaccinated adult.

**Effectiveness after the second dose**. Our estimates for direct vaccine effectiveness are consistent with the results from clinical trials and previous observational studies assessing first-dose mRNA-vaccine effectiveness. Our results document vaccine effectiveness in a real-world setting, in which it is difficult to separate the effectiveness of the first dose and the additional

protection conferred by a second dose of the vaccine (36.4% of vaccinated healthcare workers received the second dose four weeks after the first dose, Supplementary Fig. 3). However, to assess vaccine effectiveness among double vaccinated individuals, we report effectiveness estimates after the second dose using a sample that includes only double vaccinated individuals in the treatment group. Our direct effectiveness estimates are 82.7% (95% CI: 65.4% to 91.3%) 2 weeks and 82.7% (95% CI: 76.0% to 87.5%) 8 weeks after the second dose (Fig. 3a). These direct effectiveness estimates are larger after the second dose than after the first dose, as expected. Moreover, we find substantial indirect effects for unvaccinated partners: 39.1% (95% CI: −7.1% to 65.3%) 2 weeks and 39.0% (95% CI: 18.9% to 54.0%) 8 weeks after the administration of the second dose (Fig. 3b). Consistent with the results after the first vaccine dose, the indirect effects for children and adolescents aged from 3 to 18 years are statistically insignificant and smaller than for unvaccinated partners (Fig. 3c, d).

## Discussion

The present study is uniquely suited for evaluating the effect of mRNA-based vaccines on SARS-CoV-2 infections and secondary transmission from vaccinated to unvaccinated individuals. We were able to merge individual-level health records with full-population datasets containing detailed information about vaccinated individuals' occupations, partners, and children. Moreover, our research strategy capitalized on the early vaccination of healthcare workers, which created a large group of households with discordant vaccination status, critical for the estimation of household spillover effects.

Observational studies on the indirect effectiveness of mRNA-based vaccines have so far paid little attention to children[11–14]. A notable exception is a community-level study in Israel, concentrating on the regional effects of vaccinated adults on unvaccinated children's infections[15]. By contrast, here we concentrate on evaluating the effect of vaccines on transmission from vaccinated adults to unvaccinated adults and children within the same household.

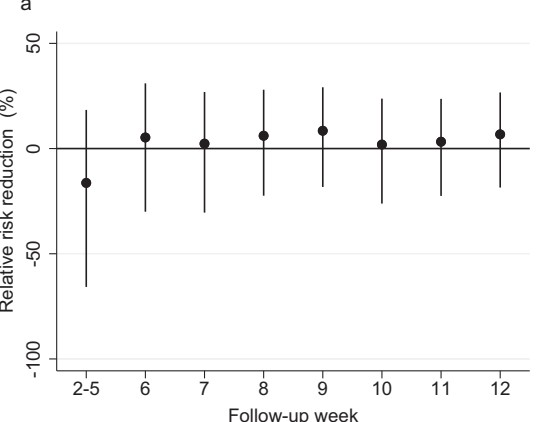
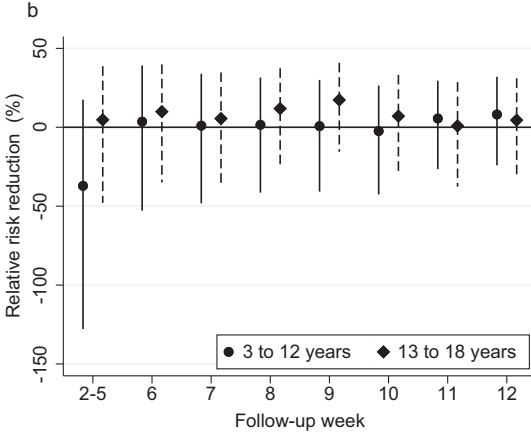

**Fig. 2 Vaccine effectiveness in unvaccinated children and adolescents.** This figure plots vaccine effectiveness estimates by week after the first dose of mRNA-based vaccination. The dependent variable is a polymerase chain reaction (PCR)-confirmed SARS-CoV-2 infection as recorded in the Finnish National Infectious Diseases Register. **a** Effectiveness estimates (relative risk reduction) for unvaccinated household members aged from 3 to 18 years who lived in the same household as the vaccinated person as of December 31, 2019. $N = 169,148$ child and adolescent observations. **b** Effectiveness estimates (relative risk reduction) separately for unvaccinated children aged from 3 to 12 years and adolescents aged from 13 to 18 years who lived in the same household as the vaccinated person as of December 31, 2019. **a** and **b** stack the estimates from follow-up weeks 2 to 5 into a single estimate as there is too few PCR-confirmed SARS-CoV-2 infections to make any inference about the short-term effectiveness of mRNA-based vaccination in this age group. $N = 105,186$ child (3–12 years) observations, 63,962 adolescent (13–18 years) observations. The estimate for weeks 2–5 represents the average vaccine effectiveness in weeks 2, 3, 4, and 5. The coefficients in both panels are estimated using a log-binomial regression model and individual-date data collapsed to individual-week data. See section Methods, Estimation samples, for details of the sample restrictions and estimation method. The relative risk reduction is presented as a percentage in data points ±95% confidence intervals. The solid black dots show relative risk reduction by week after receiving the first dose of mRNA-based vaccination. The error bars denote the 95% confidence intervals clustered at the individual level using endpoint transformation of the originally estimated confidence intervals.

Methodologically, we estimate indirect vaccine effectiveness by comparing individuals in a treatment versus a control group (defined by the vaccination status of their close contact), mimicking the RCT design. Thus, our research design complements existing observational studies that have relied on comparisons of individuals at different time periods before and after administration of a vaccine to their close contacts[11,14].

Our study has several important limitations. First, our study is limited by the fact that healthcare workers may differ from the general population, for example, in terms of their SARS-CoV-2 virus exposure. Second, virus exposure could have differed between vaccinated and unvaccinated healthcare workers as persons directly involved with COVID-19 patients were more likely to be vaccinated. These differences in virus exposure are likely to bias our vaccine effectiveness estimates towards zero. However, our results for direct effectiveness after the first and second dose are very close to the results from randomized controlled clinical trials and prior observational studies, suggesting that the sample selection and the non-random assignment of vaccines to healthcare workers do not introduce substantial bias into our direct vaccine effectiveness estimates. Third, our study is limited by the fact that we lack data on the potential biological and behavioral mechanisms that could clearly pinpoint the mechanism leading to the observed reduction in confirmed SARS-CoV-2 infections among unvaccinated adult household members. Finally, our study does not provide evidence about the risk of secondary infection conditional on a positive SARS-CoV-2 test for the index case. Thus, in contrast to a recent study in England[12], we are not able to explicitly study relative infectiousness among vaccinated and infected individuals.

Taken together, our results suggest that mRNA-based COVID-19 vaccines do not only prevent SARS-CoV-2 infections among vaccinated individuals but also lead to a substantial reduction in infections among unvaccinated adults living in the same household. However, we do not find a statistically significant decrease in the cumulative incidence of SARS-CoV-2 infections for unvaccinated children living in households with vaccinated adults. The statistical power to detect statistically significant effects in children is weakened in our study due to a smaller risk of SARS-CoV-2 infections among children than adults.

This paper provides indirect evidence that mRNA-based vaccines affect susceptibility in vaccinated individuals and may prevent transmission from vaccinated to unvaccinated adults within households. The observed reduction in household transmission from vaccinated to unvaccinated individuals is expected to curb the current COVID-19 pandemic as the household transmission of SARS-CoV-2 is believed to have a significant role in the overall spread of infections in the community[16,17].

After our study period, the SARS-CoV-2 Delta (B.1.617.2) variant has become the dominant strain in many parts of the world. While mRNA-based COVID-19 vaccines are still expected to lower the risk of infection and remain highly effective against severe disease, the emergence of the Delta variant has likely eroded direct and indirect vaccine effectiveness by increasing the likelihood of breakthrough infections and viral transmission from vaccinated individuals who become infected[18]. Further studies are required to understand whether booster vaccinations do not only improve direct vaccine effectiveness but also strengthen vaccine-associated reduction in transmission. Overall, there is a need for new evidence to understand how the indirect effects of COVID-19 vaccines on unvaccinated adults and children support the prospect of herd immunity and to inform questions related to vaccine booster strategies and the possible mass vaccination of children.

## Methods

**Data and outcome variable.** This research complies with all relevant ethical regulations. The final data used in the study were de-identified and therefore research does not constitute human subject research. Ethical approval was waived by the Institutional Review Board of the Finnish Institute for Health and Welfare (IRB: 00007085).

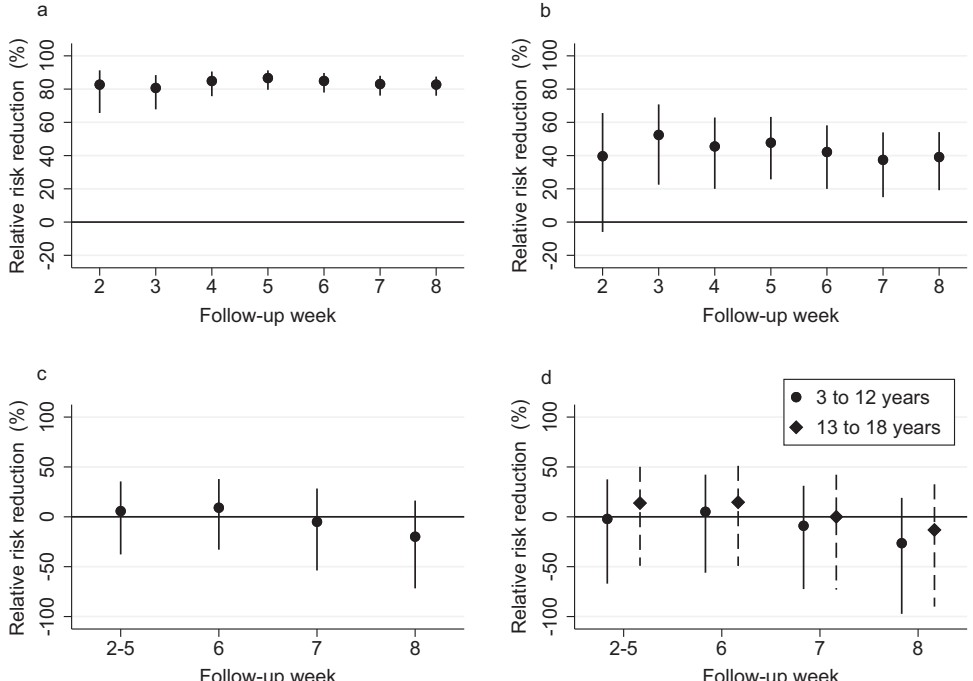

**Fig. 3 Vaccine effectiveness among fully (double) vaccinated healthcare workers and their unvaccinated partners and children living in the same household after the second vaccine dose.** This figure plots the vaccine effectiveness estimates by week after the second dose of mRNA-based vaccination. The dependent variable is a polymerase chain reaction (PCR)-confirmed SARS-CoV-2 infection as recorded in the Finnish National Infectious Diseases Register. **a** Direct effectiveness estimates (relative risk reduction) for vaccinated individuals compared to an unvaccinated control group, which is constructed by randomly assigning the beginning of a follow-up period for unvaccinated individuals. $N = 216,557$ healthcare worker observations. **b** Indirect effectiveness estimates (relative risk reduction) for the unvaccinated partners (including cohabiting partners) of fully vaccinated individuals who lived in the same household as the vaccinated person as of December 31, 2019. $N = 110,426$ partner observations. **c** Indirect effectiveness estimates (relative risk reduction) for unvaccinated household members aged from 3 to 18 years who lived in the same household as the vaccinated person as of December 31, 2019. $N = 144,935$ child and adolescent observations. **d** Effectiveness estimates (relative risk reduction) separately for unvaccinated children aged from 3 to 12 years and adolescents aged from 13 to 18 years who lived in the same household as the vaccinated person as of December 31, 2019. $N = 91,031$ child (3–12 years) observations, 53,904 adolescent (13–18 years) observations. **c**, **d** Stack the estimates from follow-up weeks 2–5 into a single estimate as there are too few PCR-confirmed SARS-CoV-2 infections to make any inference about the short-term effectiveness of mRNA-based vaccination in this age group. The estimate for weeks 2–5 represents the average vaccine effectiveness in weeks 2, 3, 4, and 5. The coefficients in all panels are estimated using a log-binomial regression model and individual-date data collapsed to individual-week data. See section Methods, Estimation samples, for details of the sample restrictions and estimation method. The relative risk reduction is presented as a percentage in data points ±95% confidence intervals. The solid black dots show relative risk reduction by week after receiving the second dose of mRNA-based vaccination. The error bars denote the 95% confidence intervals clustered at the individual level using endpoint transformation of the originally estimated confidence intervals.

The analyses were conducted using multiple population-wide administrative datasets linked at the individual level. All PCR-confirmed SARS-CoV-2 infections and their dates up to April 25, 2021, were recorded in the Finnish National Infectious Diseases Register. The Finnish National Vaccination Register provided information about the type of vaccine and the date of vaccine administration for all COVID-19 vaccines administered up to April 25, 2021. These two registers are maintained by the Finnish Institute for Health and Welfare.

We used the Finnish Incomes Register for the year 2020 from the Finnish Tax Authority to identify healthcare workers following Statistics Finland's classification of occupations. These occupations are presented in Supplementary Table 1. We merged these data with the Statistics Finland FOLK module (full-population data) for the year 2019 to identify partners (including cohabiting partners) and children who lived in the same household as the healthcare workers on December 31, 2019. FOLK is also the source for our control variables that are described in Statistical methods. Using these linked data, we constructed the outcome variable, the cumulative indicator of SARS-CoV-2 infection, for healthcare workers and their unvaccinated household members used in the statistical analysis of vaccine effectiveness.

Our linked dataset is based on nationwide administrative data sources that cover the entire Finnish population and are developed to record all SARS-CoV-2 infections and mRNA-based vaccine doses administered in Finland. However, some individuals and observations are missing in our linked dataset. The most notable missing groups of individuals are children born after 2019 and persons who immigrated to Finland after 2019, because the most recent total population data (FOLK-data) are available for the year 2019. Furthermore, our linked data do not include foreign individuals who do not have permanent residency in Finland but could have been infected during their stay in Finland. Finally, even though we use comprehensive data on the Finnish population, the household size (one of our covariates) had 1.1% and 0.2% of the total number of observations as missing for healthcare workers and their partners, respectively. Other characteristics or covariates (described in Statistical methods) had even smaller shares of missing observations. The children's sample does not have any missing observations. Individuals with missing observations in any of the covariates ($N = 2987$; 1.08%) of healthcare workers and ($N = 200$; 0.15%) of their partners are dropped from the linked dataset and therefore, from the estimation samples.

**Estimation samples.** Our analysis used three distinct estimation samples. The first sample was used in estimating the direct effectiveness of mRNA-based vaccinations (BNT162b2 by Pfizer-BioNTech or mRNA-1273 by Moderna). This sample consisted of working-age healthcare workers (aged 15–74 years). The number of healthcare workers and their vaccination status by detailed (3-digit) occupation code is shown in Supplementary Table 4. Supplementary Table 7 shows how SARS-CoV-2 infections are divided between different healthcare worker occupations. The second sample was used in estimating the indirect vaccine effect on the partners of the healthcare workers. An individual was included in this sample if their partner is a healthcare worker and they had not been vaccinated during the study period. The third sample was used in estimating the indirect vaccine spillover effect on children (aged 3–18 years) living in the same household as the healthcare worker. Supplementary Table 8 shows the number of individuals by follow-up week after the first dose in each sample. The children included in this sample are biological children of vaccinated healthcare workers. Children were included in this sample if they live in the same household as the healthcare worker and had not been vaccinated during the sample period.

**Statistical methods**. Following the current standard practice in COVID-19 vaccine effectiveness reporting, we estimate vaccine effectiveness as relative risk reduction (RRR)[19]. RRR measures how much vaccination reduced the cumulative risk of SARS-CoV-2 infection in the treatment group relative to the control group, who did not receive a vaccination. We separately investigated healthcare workers and their unvaccinated family members (partners and children).

We estimated the following log-binomial model separately for each time-to-event week $l$, defined as the number of periods in calendar week $t$ from obtaining the first (second) dose of vaccination:

$$\ln P_{it} = \alpha_l + \beta_l T_i + X_{it}\delta_l + \lambda_{tl} + e_{it} \qquad (1)$$

where $P_{it} = P\,(y_{it} = 1)$ is the probability of the cumulative SARS-CoV-2 infection of individual $i$ recorded in calendar week $t$ ($y_{it} = 1$). Moreover, $T_i$ is an indicator variable for treatment group status (vaccinated healthcare worker, partner or parent), and $\lambda_{tl}$ contains week fixed effects that capture the state of the epidemic in each time-to-event week $l$. $X_{it}$ contains controls at the individual level: age, age squared, sex, occupational group, household size (1, 2, 3, 4, 5+ household members), ethnicity (birth of origin: Finnish or foreign) and geographic area (urban, semi-urban, or rural). These covariates include the characteristics of the individuals (healthcare worker or household member) themselves, except that occupation always refers to the healthcare worker's occupation. $\alpha_l$ refers to the log risk in the control group and $\beta_l$ refers to differences in the log risks between the treatment and control groups in time-to-event week $l$. The key advantage of a log-binomial model is that it allows easy access to an estimate of the RRR: the RRR is simply $1 - \exp(\beta_l)$. The corresponding 95% confidence interval estimates were calculated using the standard errors of the $\beta_l$ coefficients. Moreover, we assessed the sensitivity of our results to non-linear (log-binomial) vs. linear estimation (linear probability model) (Supplementary Fig. 4).

As the individuals in the control group are unvaccinated, we randomly drew the beginning of the follow-up period from the observation period of vaccinations (December 27, 2020 to April 25, 2021) for these unvaccinated individuals. The random assignment of a follow-up period for each vaccinated healthcare worker and their family member parallels an assumption that vaccine coverage among healthcare workers increases linearly over time. In practice, the weekly number of vaccines administered to healthcare workers varied substantially during the study period depending on the supply of vaccines and timing of vaccine administration. Importantly, our regression models control for time-varying factors using week fixed effects, and thereby changes in the supply of vaccines and state of the epidemic over time.

**Reporting summary**. Further information on research design is available in the Nature Research Reporting Summary linked to this article.

## Data availability

The administrative healthcare and employment data used in this study are available from the Finnish Institute for Health and Welfare and Statistics Finland under restricted access due to Finnish data protection legislation. Healthcare data are also regulated under the Act on the Secondary Use of Health and Social Data (552/2019) and are, however, available upon reasonable request to and with the permission of Findata – Finnish Social and Health Data Permit Authority (https://findata.fi/en/). The Finnish Longitudinal Employer–Employee Data are available upon reasonable request to and with the permission of Statistics Finland (https://www.stat.fi). The authors are willing to assist in making data access requests.

## Code availability

All the codes (Stata 16.0) needed to reproduce the empirical results reported in this paper are available in an open science data repository GitHub at https://github.com/covidhealth/indirect-effects-covidvaccinations. The most recent release can be found at https://doi.org/10.5281/zenodo.5905898[20].

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

## Acknowledgements

M.K. is supported part by the InFLAMES Flagship Programme of the Academy of Finland (decision number: 33750). L.S. is supported part by the INVEST Flagship Programme of the Academy of Finland (decision number: 320161).

## Author contributions

Study design: J.S., M.K., T.S., and L.S. Data collection: J.S., M.H., and L.S. Data analysis: J.S., M.H., and L.S. Data interpretation: J.S., M.H., M.K., T.L., T.S., M.S., and L.S. Writing: J.S., M.H., M.K., T.S., M.S., and L.S.

## Competing interests

M.K. declares to his employer a grant, but no personal support or financial relationship, from Pfizer during the conduct of the study. The other authors have no competing interest to declare.
