## [Peer Review File · Nature Communications]

The indirect effect of mRNA-based COVID-19 vaccination on healthcare workers' unvaccinated household membersREVIEWER COMMENTS

Reviewer #1 (Remarks to the Author):

This is a very interesting paper and the results are presented clearly.

1. It would be great to see some descriptive tables summarising the demographics between vaccinated and unvaccinated healthcare workers (as well as their household members). I understand this is briefly mentioned in the appendix but more details should be added.
2. More importantly, it would be good to show the number of SARS-CoV-2 infections in these groups when making the comparison and estimation on vaccine effect. This is particularly essential considering that there might be a small number of positive cases in some subgroup analysis which would affect the accuracy of the results.
3. There is no mentioning of missing data which are common when using routinely collected linked datasets.
4. There is no clear explanation of why the authors only looked at the results from two weeks onwards. What about the first two weeks?
5. The study period is until 24th March 2021. I wonder if the authors can update the analysis until a more recent date which would give them a larger sample size with more statistical power.
6. The population under study are healthcare workers and their household members, which should be highlighted in the title and abstract.
7. The reduction showed among children and young adults under 18 years is not substantial (or very limited), so the conclusion needs to be changed.
8. There are quite a few supplement results listed in Appendix but they were not mentioned in the paper. What are their purposes?
9. What about second dose vaccine effect?
10. Do you look at the COVID-19 hospital admission?

Reviewer #2 (Remarks to the Author):

The manuscript by Salo et al presents estimates of direct and indirect effectiveness of mRNA-based Covid-19 vaccines among vaccinated healthcare workers (HCW) and their unvaccinated household contacts. The estimates of direct and indirect effectiveness are based on the cumulative risk of infection among vaccinated vs unvaccinated HCWs and their household contacts, respectively, and are provided by week of follow-up after the first dose of the vaccine. While understanding the indirect effects of Covid-19 vaccines in particular is of interest right now, clarification around how to interpret the indirect effectiveness estimates and why they vary by week of follow-up, as well as some additional details on the data used, is needed.

Major comments:

1. My first concern is that it is not clear how to interpret the change in direct (DE) and indirect (IE) effectiveness estimates by week of follow-up. For the DE estimates, this is like due at least in part to the additional protection conferred by a second dose of the vaccine, which based on the

information in Figure A1, was administered to ~40% of HCWs 3-4 weeks after the first dose; presumably (based on data from the clinical trials), full immunity from the second dose would take another 1-2 weeks. However, that does not explain why the DE estimates continue to increase between 6-10 weeks of follow-up. What is the potential explanation(s) for this increase in DE estimates, and are the DE estimates expected to plateau at some point?

2. For the IE estimates, it is more reasonable to expect that these estimates should vary based on the duration of follow-up. The indirect effectiveness estimates incorporate both the relative risk that a vaccinated HCW is infected and that they pass on the infection to their unvaccinated spouse or child/children. Thus, I would expect the estimates to be closer to the null 2 weeks after vaccination of the HCW, since many of these secondary infections among unvaccinated household members could represent transmission that occurred before the HCW was vaccinated (or protected by the vaccine) and/or transmission that occurred from outside the household (which should not vary depending on the vaccination status of the spouse/parent). Unlike the DE estimates, the IE estimates seem to stabilize/plateau after 8 weeks of follow-up.

3. In the Discussion, the authors should clarify that this study does not explicitly estimate the relative infectiousness given infection of vaccinated HCWs (e.g. similar to the recent study of UK HCWs by Harris et al); instead, it provides indirect evidence of there being an effect of vaccination on transmission. It could be useful to highlight, for example among the limitations, that the authors are not conditioning on their being an infected index case in the household or explicitly accounting for the risk of transmission from other household members or from outside the household (as noted above).

4. For the model estimating the IE for children, it seems as though the control for potential confounding by age and sex is based only on the age and sex of the oldest child (bottom of p. 6 of the supplement). Is this correct? If so, there may be considerable residual confounding by age (and sex) for households with multiple children. There also seem to be more young children per household amongst the unvaccinated households compared to the vaccinated households (based on Table A3). Is this correct? If so, it could further bias the analysis.

5. In the main text, there is no clear presentation of the data: for example, how many individuals were followed, how many households, vaccination coverage among HCWs, and number of SARS-CoV-2 infections in the study. These numbers are presented in the tables in the supplementary appendix, but should also be mentioned in the main text. Also, the follow-up period should explicitly be referred to in the main text.

6. What is the proportion of fully vaccinated individuals during the study period (i.e., December 27, 2020 - March 18, 2021)? Is it possible to include an additional analysis restricted to fully vaccinated individuals and/or extended to a time period when most HCWs were fully vaccinated?

7. There seems to be quite a bit of variation in vaccine coverage depending on the occupational code of the HCW (according to Table A4). It might be interesting to examine whether this is any variation in the DE and IE estimates depending on the occupational code. Also, is the HCW definition the same as the designations that determined early eligibility for the vaccine?

Minor comments:

8. Lines 58-59: Isn't the analysis limited to individuals without a history of previous infection? If so, it is not clear to me why there should be selection bias.

9. Lines 92-93: By "treated", do you mean "vaccinated"? It would be useful to clarify. There is also a typo in this sentence (delete "of").

10. Lines 95-97: It may be worth noting that the likely direction of bias would be towards the null.

11. Figures A2-A7: It would be useful to present the estimates from the main analysis alongside those from the sensitivity analyses (i.e. in the same plot, perhaps in a different color) in order to more easily compare.

12. Figure A6-A7: For this analysis, is the follow-up period for unvaccinated households always December 27 to January 10 (for 2 weeks of follow-up), January 17 (for 3 weeks of follow-up), etc? If so, that should be clarified. Or this analysis should be deleted, since there is a high potential for bias based on the state of the epidemic at different time points.

13. The analysis code should be made available in a public repository (e.g. github).

Reviewer #3 (Remarks to the Author):

This is an important study looking at the indirect effectiveness of vaccines reducing transmission in households. The authors developed a really nice controlled study, adjusting for age, sex and time of the epidemic, and demonstrates significant transmission reduction in unvaccinated household members.

I have a few comments to improve the study and presentation of findings:

1- It is well established that large households have 2-3 times higher secondary attack rates. This would be an important factor when looking at indirect transmission reduction. While the effect of household size will be adjusted when deprivation is taken into account, this analysis does not take deprivation into account. We are assuming all these families have similar risk of infection.

2- There are several studies showing reduction in infection and transmission in other countries, especially data from the UK supports this observation, it would be good to acknowledge available data: <https://academic.oup.com/ofid/advance-article/doi/10.1093/ofid/ofab259/6278371>

3- Another important aspect is that we are assuming all HCWs have the same risk of exposure at work, but some will be much more at risk than others. For instance, working in community setting might be more risk than secondary care. Is there any possibility to take this into account? And do we know which type of mitigations were in place at work for these HCWs? Is there any difference between cases and controls?

4- Another confounder in the risk of infection could be ethnicity. Even among HCWs, we know that ethnic minority groups including HCWs are at higher risk of infection, and there is a risk that they may be less likely to vaccinated due to religious reasons. Therefore, when adjusting for risk between cases/controls this may be an important confounder.

Dear Reviewer,

We would like to thank you for your detailed review report and constructive comments that have helped us improve the quality of our manuscript. We were pleased to read that you find our manuscript interesting and clearly presented. Please find a detailed point-by-point response to your concerns and remarks below. Your comments are shown in bold font. Our responses are written using regular font.

Reviewer #1 (Remarks to the Author):

This is a very interesting paper and the results are presented clearly.

1. It would be great to see some descriptive tables summarising the demographics between vaccinated and unvaccinated healthcare workers (as well as their household members). I understand this is briefly mentioned in the appendix but more details should be added.

Thank you for your suggestion to add more details to descriptive tables summarizing the demographics between vaccinated and unvaccinated healthcare workers. To address this suggestion, we provide in the revised manuscript more details about the individuals' household size, geographic area (urban, semi-urban, rural), occupational group, and ethnicity. We construct the variable for ethnicity using the classification available in the official statistics of Finland: (i) Finnish background (all persons with at least one parent born in Finland), (ii) Foreign background (persons whose both parents or the only known parent were born abroad). We summarize this demographic information in the main text (Introduction, paragraph 5) and report demographic information in more detail in the Supplementary Information (Supplementary Tables 2 and 3). Importantly, we have also added these additional demographic variables to our regression models and find that our estimates are largely robust to the inclusion of these additional control variables.

2. More importantly, it would be good to show the number of SARS-CoV-2 infections in these groups when making the comparison and estimation on vaccine effect. This is particularly essential considering that there might be a small number of positive cases in some subgroup analysis which would affect the accuracy of the results.

We agree that it is important to show the number of SARS-CoV-2 infections in our treatment and control groups. To address this comment, we now report the number of infections in different estimation samples in the main text and the number of infections for all follow-up weeks in Supplementary Figure 1. It is also worth emphasizing that our vaccine effectiveness estimates are based

on the cumulative incidence of SARS-CoV-2 infections, which are reported in Supplementary Figure 2. Thus, the regression results we report in our revised manuscript are based on the cumulative (absolute) incidence of infections in different groups by follow-up week.

3. There is no mentioning of missing data which are common when using routinely collected linked datasets.

We agree that it is useful to mention the most important potential sources of missing data. Our study is based on nationwide administrative registers (Finnish Infections Diseases Register, Finnish National Vaccine Register, the Incomes Register and Statistics Finland FOLK-data, a dataset that covers the entire Finnish population) that are designed to record all SARS-CoV-2 infections and mRNA-based vaccine doses administered in Finland. However, as mentioned, linking these registers may lead to missing data in some instances. The most notable missing groups of individuals in our linked data are children born after year 2019 and persons who have immigrated to Finland after year 2019, because the most recent total population data are available for year 2019. Thus, we cannot link young children to their parents and do not observe occupation codes and other demographic characteristics for individuals who have immigrated to Finland after year 2019. A further limitation of our linked dataset is that it includes information only about individuals who have a permanent residency in Finland. Consequently, we are missing occupation codes and demographic information for foreign health care workers who are commuting to work in Finland.

We summarize in the revised manuscript the most important potential sources of missing data and write (in the Methods section) that “The most notable missing groups of individuals in our linked data are children born after year 2019 and persons who have immigrated to Finland after year 2019, because the most recent total population data are available for year 2019. Furthermore, our linked dataset does not include foreign individuals who do not have a permanent residency in Finland but could have been infected during their stay in Finland.”

4. There is no clear explanation of why the authors only looked at the results from two weeks onwards. What about the first two weeks?

Thank you for mentioning the lack of explanation for the practice of reporting the results from two weeks onward. To address this remark, we write in the revised manuscript that “We report vaccine effectiveness estimates from follow-up week two onwards because the cumulative incidence of infections is very small in the first two follow-up weeks in our sample of healthcare workers and their household members (Supplementary Figure 2).” It is also worth mentioning that the serological responses after the first vaccine doses are fairly limited during the first weeks and the biological

mechanism of vaccines is therefore questionable. Moreover, the healthy-vaccinated effect might distort the effect during the first days after vaccination.

5. The study period is until 24th March 2021. I wonder if the authors can update the analysis until a more recent date which would give them a larger sample size with more statistical power.

Thank you for pointing out the benefit of updating the analysis until a more recent date. We agree that extending the study period is useful in many ways and have extended the study period until 25th April 2021 (at the time of first submission we had data only until 24th March 2021 but have now access to an updated dataset). This date marks the beginning of Covid-19 vaccine administration in general adult population (individuals not belonging to any medically defined risk group or working in the health care sector) aged 69 years and below in Finland.

Extending the analysis until the 25th April has the following consequences on our analysis: (i) it increases the share of vaccinated health care workers in our sample (now the share of healthcare workers with at least one dose of mRNA-based vaccines is 42%), (ii) it enables longer follow-up time, and (iii) makes sure that there remains enough unvaccinated healthcare workers and household members in the sample (extending the study period substantially beyond the 25th April would mean that the vast majority of individuals in our sample would be vaccinated, leaving no room to study indirect vaccine effectiveness). However, it is important to notice that extending the study period beyond 24th March 2021 does not lead to a larger overall sample size, because our sample previously already included all healthcare workers that met our inclusion criteria. In fact, extending the analysis until the 25th April led to a somewhat smaller total sample size, because a small proportion of healthcare workers were vaccinated using the ChAdOx1 nCoV-19 (Astra Zeneca) vaccines after 24th March and have been excluded from our new samples and analyses.

6. The population under study are healthcare workers and their household members, which should be highlighted in the title and abstract.

We now explicitly mention the population under study (healthcare workers and their household members) in the title and abstract.

7. The reduction showed among children and young adults under 18 years is not substantial (or very limited), so the conclusion needs to be changed.

We agree. Moreover, the results based on updated data and analyses do not indicate statistically significant effects among children and young adults under 18 years even at the end of the follow-up period (12 weeks). We have revised the text and conclusions accordingly.

8. There are quite a few supplement results listed in Appendix but they were not mentioned in the paper. What are their purposes?

Thank you for pointing out the large number of additional results in the Appendix and not clearly mentioning the purpose of these results. The purpose of these analyses is to study the robustness of our main model specification to certain alternative estimation strategies and techniques. In the revised version of the paper we study the sensitivity of our results to (i) non-linear (log-binomial) vs. linear estimation and (ii) the definition of control group using randomly assigned beginning of a follow-up period (in control group) vs. using the beginning of the mass vaccination program (December 27, 2020).

We agree that reporting all these robustness checks could overwhelm several potential readers. However, we also feel that the supplemental results listed above are reasonably important to evaluate to robustness of our main model specifications and have decided to report these two sensitivity analyses in the Appendix. We now mention in the revised manuscript more explicitly the purpose of these supplemental results and write (in Methods section) that: “We also conducted a sensitivity check, where we followed individuals in the control group from the start of the mass vaccination program, December 27, 2020 (Supplementary Figure 4). Moreover, we assessed the sensitivity of our results to non-linear (log-binomial) vs. linear estimation (linear probability model) (Supplementary Figure 5).”

9. What about second dose vaccine effect?

Thank you for raising an important point about the second dose vaccine effect. To address this question and related remarks by reviewer #2, we have conducted additional analyses that are restricted to fully (twice) vaccinated individuals. These additional analyses enable us to study direct and indirect vaccine effectiveness among fully vaccinated individuals after the second vaccine dose. These new analyses are reported in Figure 3.

Consistent with the existing literature, we find that direct and indirect vaccine effectiveness estimates are larger in a sample that is restricted to fully vaccinated individuals than in a sample that contains once and fully vaccinated individuals. Our additional analyses restricted to fully vaccinated individuals

suggest that the second vaccine dose does not only improve direct vaccine effectiveness but increases the indirect vaccine effectiveness for adults. However, it is important to notice that comparing vaccine effectiveness after the administration of first and second vaccine dose does not enable unbiased inference about the marginal effect of second vaccination dose in our research design, as the second dose was administered only to individuals who were not infected after the first dose. Thus, a simple comparison of once and twice vaccinated individuals could be subject to a notable selection bias.

10. Do you look at the COVID-19 hospital admission?

In this paper, we are not looking at the effects on COVID-19 related hospital admissions for two reasons. First, there is not enough observations (hospital admissions) to reliably evaluate the indirect effects, especially on children. There are in total approximately 20 admissions for the groups of healthcare workers and their spouses and less than 5 hospital admissions for children during the follow-up. Secondly, the data on hospital admissions in the Infectious Disease Register are somewhat sensitive to measurement error due to insufficient reporting of admissions by some hospitals and do not necessarily include all COVID-19 related hospital admissions.

Dear Reviewer,

We would like to thank you for your detailed review report and constructive comments that have helped us to improve the quality of our manuscript. We were pleased to read that you find our manuscript interesting and timely. Please find a detailed point-by-point response to your concerns and remarks below. Your comments are shown in bold font. Our responses are written using regular font.

Reviewer #2 (Remarks to the Author):

The manuscript by Salo et al presents estimates of direct and indirect effectiveness of mRNA-based Covid-19 vaccines among vaccinated healthcare workers (HCW) and their unvaccinated household contacts. The estimates of direct and indirect effectiveness are based on the cumulative risk of infection among vaccinated vs unvaccinated HCWs and their household contacts, respectively, and are provided by week of follow-up after the first dose of the vaccine. While understanding the indirect effects of Covid-19 vaccines in particular is of interest right now, clarification around how to interpret the indirect effectiveness estimates and why they vary by week of follow-up, as well as some additional details on the data used, is needed.

Thank you for the precise summary of our paper. We hope that the revised version of our manuscript clarifies the interpretation of indirect effectiveness estimates and other questions raised in your comments.

Major comments:

1. My first concern is that it is not clear how to interpret the change in direct (DE) and indirect (IE) effectiveness estimates by week of follow-up. For the DE estimates, this is like due at least in part to the additional protection conferred by a second dose of the vaccine, which based on the information in Figure A1, was administered to ~40% of HCWs 3-4 weeks after the first dose; presumably (based on data from the clinical trials), full immunity from the second dose would take another 1-2 weeks. However, that does not explain why the DE estimates continue to increase between 6-10 weeks of follow-up. What is the potential explanation(s) for this increase in DE estimates, and are the DE estimates expected to plateau at some point?

We agree that our observational study design does not allow clearly separate the effectiveness of the first dose and the additional protection conferred by a second dose of the vaccine. However, we provide in the revised manuscript results from new analyses that help to interpret the changes in effectiveness estimates by week of follow-up. First, we have extended the follow-up period using newly available data until April 25, 2021, that marks the beginning of Covid-19 vaccine administration in general adult population in Finland. Second, we provide additional analyses that are restricted to fully vaccinated individuals. As shown in (revised) Figure 1, extending the time period led to more stable effectiveness estimates that appear to plateau around follow-up week 8. While our data does not allow us to study the mechanisms behind the effectiveness estimates in detail, the second dose of the vaccine can at least partly explain why DE estimates increase until week 8 or so. Moreover, it is important to note that the changes in DE estimates for 6-9 weeks are small.

2. For the IE estimates, it is more reasonable to expect that these estimates should vary based on the duration of follow-up. The indirect effectiveness estimates incorporate both the relative risk that a vaccinated HCW is infected and that they pass on the infection to their unvaccinated spouse or child/children. Thus, I would expect the estimates to be closer to the null 2 weeks after vaccination of the HCW, since many of these secondary infections among unvaccinated household members could represent transmission that occurred before the HCW was vaccinated (or protected by the vaccine) and/or transmission that occurred from outside the household (which should not vary depending on the vaccination status of the spouse/parent). Unlike the DE estimates, the IE estimates seem to stabilize/plateau after 8 weeks of follow-up.

We agree that the IE estimates incorporate the relative risk of being infected after vaccination and passing the infection on unvaccinated spouse or children. Thus, we would expect that the IE estimates are close to null after 2 weeks and potentially plateau later than the DE estimates. Consistently with the mentioned expectations, our revised analyses show smaller and insignificant IE estimates for follow-up weeks 2-5 and significant estimates after 6 weeks of follow-up. Using our new extended dataset, we find that the IE estimates plateau slightly before the DE estimates. While our data do not allow us to study the specific mechanisms potentially driving these results, the second dose effects (reported in Figure 3) could potentially explain these findings.

3. In the Discussion, the authors should clarify that this study does not explicitly estimate the relative infectiousness given infection of vaccinated HCWs (e.g. similar to the recent study of UK HCWs by Harris et al); instead, it provides indirect evidence of there being an effect of vaccination on transmission. It could be useful to highlight, for example among the

limitations, that the authors are not conditioning on their being an infected index case in the household or explicitly accounting for the risk of transmission from other household members or from outside the household (as noted above).

Thank you for pointing out a need to clarify that our study does not explicitly estimate the relative infectiousness given infections of vaccinated individuals (e.g. similar to Harris et. al 2021). We agree that there is a clear distinction between studies that provide indirect evidence about the effect of vaccination on transmission and studies that are conditioning estimates on being infected. We explicitly note this distinction in the revised manuscript and write in the Discussion section (limitations) that: “Finally, our study does not provide evidence about the risk of secondary infection conditional on positive SARS-CoV-2 test for the index case. Thus, in contrast to a recent study from England (Harris et al. 2021), we are not able to explicitly study relative infectiousness among vaccinated and infected individuals.”

4. For the model estimating the IE for children, it seems as though the control for potential confounding by age and sex is based only on the age and sex of the oldest child (bottom of p. 6 of the supplement). Is this correct? If so, there may be considerable residual confounding by age (and sex) for households with multiple children. There also seem to be more young children per household amongst the unvaccinated households compared to the vaccinated households (based on Table A3). Is this correct? If so, it could further bias the analysis.

We agree that in the original manuscript the results for children were crude household-level estimates of vaccine effectiveness. It is correct that models controlled for potential confounding in the children sample only using the age and sex of the oldest child. To address this concern and other related remarks, we have linked several new demographic variables to the dataset. Our new dataset includes demographic variables for children (including sex and age) at the individual level. To provide more robust estimates of vaccine effectiveness among children, our revised models for children control for sex and age at the individual level. Furthermore, we have added additional demographic variables (household size, ethnicity and geographic area) to all regression models and find that our estimates are robust to the inclusion of these additional control variables.

5. In the main text, there is no clear presentation of the data: for example, how many individuals were followed, how many households, vaccination coverage among HCWs, and number of SARS-CoV-2 infections in the study. These numbers are presented in the tables in the supplementary appendix, but should also be mentioned in the main text. Also, the follow-up period should explicitly be referred to in the main text.

Thank you for the suggestion to add more descriptive statistics to the main text. To address this remark, we summarize in the revised manuscript key demographic information in the main text (Introduction, paragraph 5). In addition, following the suggestion by Reviewer #1, we now provide more information about individuals' ethnicity, household size and geographic area. This new demographic information is summarized in the Supplementary materials (Supplementary Tables 2 and 3).

6. What is the proportion of fully vaccinated individuals during the study period (i.e., December 27, 2020 - March 18, 2021)? Is it possible to include an additional analysis restricted to fully vaccinated individuals and/or extended to a time period when most HCWs were fully vaccinated?

Thank you for raising an important question about vaccine effectiveness among fully (twice) vaccinated individuals. To address this question, we have conducted additional analyses that are restricted to fully vaccinated individuals. These additional analyses enable to study direct and indirect vaccine effectiveness among fully vaccinated individuals after the second dose of the vaccine. These new analyses are reported in Figure 3, a-d. At the same time, we have extended the study period until 25th April 2021 (at the time of first submission we had data only until 24th March 2021 but have now access to an updated dataset). This date marks the beginning of Covid-19 vaccine administration in general adult population (individuals not belonging to any medically defined risk group or working in the health care sector) in Finland.

Consistent with the existing literature, we find that the DE and EI estimates are larger in a sample that is restricted to fully vaccinated individuals than in a sample that contains once and fully vaccinated individuals. Our additional analyses restricted to fully vaccinated individuals suggest that the second vaccine dose does not only improve direct vaccine effectiveness but increases the indirect vaccine effectiveness for adults. However, it is important to notice that comparing vaccine effectiveness after the administration of first and second vaccine dose does not enable unbiased inference about the marginal effect of second vaccination dose in our research design, as the second dose was administered (in Finland) only to individuals who were not infected after the first dose. Thus, a simple comparison of once and twice vaccinated individuals is subject to a selection bias.

7. There seems to be quite a bit of variation in vaccine coverage depending on the occupational code of the HCW (according to Table A4). It might be interesting to examine whether this is any variation in the DE and IE estimates depending on the occupational code.

Also, is the HCW definition the same as the designations that determined early eligibility for the vaccine?

While there is indeed some variation in the share of vaccinated healthcare workers among considered occupations, the share of vaccinated is not very low in any single occupational group. Moreover, it is important to emphasize that our results are not sensitive for including/excluding occupational group indicator variables. We agree that it would be also interesting to examine whether DE and IE estimates vary depending on the occupational code. However, we decided to leave this for future research, as some of the occupation groups are relatively small in our sample of healthcare workers.

Minor comments:

8. Lines 58-59: Isn't the analysis limited to individuals without a history of previous infection? If so, it is not clear to me why there should be selection bias.

Yes, the analysis is limited to individuals without history of previous infection. We agree that the sentence in the previous of our manuscript on lines 58-59 was potentially misleading. We were referring to the estimation of second vaccine dose that could be affected by selection bias, as only those without infection after the first dose were able to get the second dose.

9. Lines 92-93: By "treated", do you mean "vaccinated"? It would be useful to clarify. There is also a typo in this sentence (delete "of").

We have changed the text as suggested.

10. Lines 95-97: It may be worth noting that the likely direction of bias would be towards the null.

We have changed the text as suggested.

11. Figures A2-A7: It would be useful to present the estimates from the main analysis alongside those from the sensitivity analyses (i.e. in the same plot, perhaps in a different color) in order to more easily compare.

Thank for the suggestion to present the results from the main analysis and sensitivity analyses side-by-side in the same figure. We have changed the presentation of sensitivity analyses as suggested.

12. Figure A6-A7: For this analysis, is the follow-up period for unvaccinated households always December 27 to January 10 (for 2 weeks of follow-up), January 17 (for 3 weeks of follow-up),

etc? If so, that should be clarified. Or this analysis should be deleted, since there is a high potential for bias based on the state of the epidemic at different time points.

This is correct interpretation and we have clarified discussion regarding this alternative statistical specification in the supplementary analysis. We admit that the state of the epidemic could affect results in this specification, but have decided to report these estimates in the supplementary material (despite the potential flaws) for comparison as similar type of analysis have been conducted in some other observational studies on vaccine effectiveness. It is also worth emphasizing that the results from this specification are not very different from the results of our main specification using a random follow-up date as an index event in the control group.

13. The analysis code should be made available in a public repository (e.g. github).

We are committed to publish the analysis code in a public repository (github) once the paper is published.

Dear Reviewer,

We would like to thank you for your detailed review report and constructive comments that have helped us improve the quality of our manuscript. We were pleased to read that you find our study important and carefully conducted. Please find a detailed point-by-point response to your concerns and remarks below. Your comments are shown in bold font. Our responses are written using regular font.

Reviewer #3 (Remarks to the Author):

This is an important study looking at the indirect effectiveness of vaccines reducing transmission in households. The authors developed a really nice controlled study, adjusting for age, sex and time of the epidemic, and demonstrates significant transmission reduction in unvaccinated household members.

Thank you for these kind words. This is much appreciated in these busy times.

I have a few comments to improve the study and presentation of findings:

1- It is well established that large households have 2-3 times higher secondary attack rates. This would be an important factor when looking at indirect transmission reduction. While the effect of household size will be adjusted when deprivation is taken into account, this analysis does not take deprivation into account. We are assuming all these families have similar risk of infection.

Thank you for pointing out the role of household size in secondary attack rate. We agree that it is important to take household size into account when looking at the indirect transmission reduction. Using an updated dataset, we have been able to include new demographic variables in our analysis and find that the probability of infection is indeed higher in large households. To address this concern when looking at the indirect transmission, we have now added household size variable (along with other new controls) to our regression models and find that our indirect vaccine effectiveness estimates are largely robust to the controlling for household size, even though the new point estimates for indirect effectiveness are somewhat smaller than in our original manuscript.

2- There are several studies showing reduction in infection and transmission in other countries, especially data from the UK supports this observation, it would be good to acknowledge available data: <https://academic.oup.com/ofid/advance-article/doi/10.1093/ofid/ofab259/6278371>

Thank you for bringing the paper by Richterman et al. (2021) into our attention. We have now carefully examined the paper. Consequently, we have updated our references and aspire to acknowledge all relevant available evidence as discussed by Richterman et al. (2021).

3- Another important aspect is that we are assuming all HCWs have the same risk of exposure at work, but some will be much more at risk than others. For instance, working in community setting might be more risk than secondary care. Is there any possibility to take this into account? And do we know which type of mitigations were in place at work for these HCWs? is there any difference between cases and controls?

Unfortunately, we do not have detailed information where HCWs are working (e.g. community vs. secondary care setting) or whether they face patients in their work or not. In any case we have now included indicator variables for different type of HCWs (occupation codes) in our regression models. While there is some variation in the share of vaccinated workers among considered occupations, the share of vaccinated is not very low in any single occupation. Moreover, it is important to emphasize that our results are not sensitive for including/excluding these indicator variables. Regarding mitigation practices, we do not have detailed data on mitigation practices in place or especially how they could potentially differ among treatment and control groups.

4- Another confounder in the risk of infection could be ethnicity. Even among HCWs, we know that ethnic minority groups including HCWs are at higher risk of infection, and there is a risk that they may be less likely to vaccinated due to religious reasons. Therefore, when adjusting for risk between cases/controls this may be an important confounder.

We agree that ethnicity is expected to be associated with the risk of infection potentially even among HCWs. To address this and related concerns, we have linked several new demographic variables to the dataset. Our new linked dataset now includes a variable for individual's ethnicity at the individual level using the classification available in the official statistics of Finland: (i) Finnish background (all persons with at least one parent born in Finland), (ii) Foreign background (persons whose both parents or the only known parent were born abroad). To provide more robust estimates of vaccine effectiveness, our revised models control for ethnicity using the available categorical variable for ethnicity. We find that

our estimates are largely robust to controlling for ethnicity, even though the new point estimates for indirect effectiveness are somewhat smaller than in our original manuscript.

REVIEWER COMMENTS

Reviewer #2 (Remarks to the Author):

While I appreciated the efforts that the authors made to improve the manuscript, I still have some major concerns about the interpretation of the results and the potential for confounding in the analysis.

Major comments:

1. In my first review, I expressed concern about changes in the direct effectiveness (DE) estimates increasing between weeks 6 to 10 with no apparent explanation. In their response, the authors state that by increasing the duration of the study until April 25, DE estimates seem to plateau around week 8. Also, they highlight that by adding a separate analysis for the DE following the second dose of the vaccine partially explains this increase. However, I think there still needs to be some discussion added to the text explaining why the DE estimates increase between weeks 2-8. Likewise, I would expect some discussion around the indirect effectiveness (IE) estimates any why they should be expected to increase over time.

2. Importantly, the authors make no mention of how these results might change in light of the emergence of the Delta variant. It is possible that the results could look very different if taking into account the recent period when the Delta variant has become dominant. While I understand that further extending the analysis would not be possible (i.e. the HCWs would all be vaccinated), the authors should explicitly discuss how the emergence of the Delta variant might affect both the DE and IE estimates in the Discussion.

3. Reviewer 1 asked for the number of infections broken down by subgroup, which could help to indicate the potential for confounding by the covariates included in the analysis. Including the total number of infections in the vaccinated vs unvaccinated groups is not really addressing the reviewer's suggestion.

4. I am concerned about the causal language in the abstract where the authors write: "here we show that mRNA-based Covid-19 vaccines do not only protect vaccinated individuals..". It is possible the results could be due to confounding. Weaker language (e.g. "vaccination is associated with...") should be used.

5. As a sensitivity analysis, the authors assigned the start of the follow-up period with the start of the vaccination campaign. As pointed out in my previous review, this analysis has inherent bias due to the state of the epidemic over time. The authors reply that this analysis has been conducted in other studies, which is not a valid reason in my opinion (and I'm not aware of any such studies), and no references are provided. Also, they emphasize that the results from the sensitivity analysis are in agreement with the main analysis where a random follow-up date is used. I do not find this very reassuring, as it could represent evidence of potential confounding in the main analysis.

6. To explain the non-significant indirect effectiveness estimate for children, the authors say this may be due to "the notion that children could be less susceptible to SARS- CoV-2 from other household members than adults". I don't feel that this interpretation is necessarily correct. Why should this influence the RELATIVE risk for children living with a vaccinated vs unvaccinated HCW? I would think it should only affect the power to detect a significant effect, not the point estimate itself. Further explanation is needed.

7. The authors include a description of their data in the Introduction. While this is important information to include in the main text, I don't think it's appropriate to include it in the Introduction. Either the manuscript should be reformatted to include the Methods before the Results section, or the essential methods (included in the last few paragraphs of the Introduction) should be incorporated into the Results section.

8. In the Discussion, the authors refer to “existing observational studies that have relied on comparisons of vaccinated individuals at different time periods before and after administration of a vaccine to their close contact” but do not provide any reference to these studies.

9. By randomly assigning a follow-up period for each unvaccinated HCW and their family members in the control group, I think the authors are inherently assuming that the vaccine coverage among HCWs increases approximately linearly during the study period, which is not necessarily the case here.

Minor comments:

Figure S1: The authors should clarify in the figure legend why the total cumulative number of infections decreases (i.e. it reflects the corresponding decrease in the number of individuals). Otherwise, the figure could be misleading.

Figure S2: Y-axis should specify that this is a % of the population.

Lines 155: the lower 95%CI is missing.

Line 300: What do the authors mean by: “recorded in calendar week t and thereafter”?

Line 302: Need to clarify that they do not control for characteristics of both the HCW and their contact.

Figure 2: Infections are combined between weeks 2-5 since the numbers are low. How does this differ from the cumulative estimate for 5 weeks of follow-up? In other words, do the estimates for week 6 only include infections between weeks 5-6?

Lines 43-44: “Evidence that vaccinated individuals infected with SARS-CoV-2 have lower viral loads” might not be true for the circulating Delta variant.

Reviewer #3 (Remarks to the Author):

Many thanks for the revised manuscript, my comments are addressed.

Dear Reviewer,

We would like to thank you again for your very detailed review report and constructive comments that have helped us to further improve the quality of our manuscript. Please find a detailed point-by-point response to your concerns and remarks below. Your comments are shown in bold font. Our responses are written using regular font.

Reviewer #2 (Remarks to the Author):

While I appreciated the efforts that the authors made to improve the manuscript, I still have some major concerns about the interpretation of the results and the potential for confounding in the analysis.

Major comments:

1. In my first review, I expressed concern about changes in the direct effectiveness (DE) estimates increasing between weeks 6 to 10 with no apparent explanation. In their response, the authors state that by increasing the duration of the study until April 25, DE estimates seem to plateau around week 8. Also, they highlight that by adding a separate analysis for the DE following the second dose of the vaccine partially explains this increase. However, I think there still needs to be some discussion added to the text explaining why the DE estimates increase between weeks 2-8. Likewise, I would expect some discussion around the indirect effectiveness (IE) estimates any why they should be expected to increase over time.

Thank you for the suggestion to extend the discussion on the increase of DE estimates between weeks 2 – 8. Compared to the existing evidence from several other observational studies, we observe delayed plateauing of direct vaccine effectiveness estimates. We do not have a definite explanation why the DE estimates increase between weeks 2 – 8. This tendency might be explained by several factors as discussed below. Notably, we do not observe increasing vaccine effectiveness over time after the second dose (DE and IE estimates stabilize roughly two weeks after the second dose). In our view, this suggest that there is no systematic bias that would lead to increasing effectiveness estimates over time.

Overall, the plateauing of DE estimates around week 8 could be explained by several factors. First, it takes time to develop the full immune response to the vaccine (related observational studies have not reported increasing vaccine effectiveness until follow-up week 8, but at least two observational papers have reported increasing direct vaccine effectiveness after single vaccine dose up to 34 days (Bernal et al. 2021 and Shrotri et al. 2021). Second, and most importantly, our DE estimates (Fig 1a) include a mixture of individuals who have received either one or two vaccine doses. The additional protection conferred by the second vaccine dose is expected to be observed, at the earliest, 4 to 5 weeks after the beginning of the follow-up (there was a minimum of three weeks between the first and second dose). Third, the plateauing DE vaccine estimates might be explained by several contextual differences between the previously reported results and our findings including, among others, different testing strategies (more intensive testing strategies are expected to lead to earlier detection of infections), study populations and differences in administrative health care register protocols.

The temporal pattern of indirect vaccine effectiveness estimates is expected to roughly follow the pattern of direct vaccine effectiveness estimates. Consequently, the same factors that might explain the temporal pattern of DE estimates are expected to explain the pattern of IE estimates. The most likely explanation for the increasing IE estimates over time after the first dose is the additional protection conferred by the second vaccine dose, as discussed above. Based on comparison of DE and IE estimates after the first and second dose, there are, in our view, no major discrepancies between the temporal pattern of DE and IE estimates.

Overall, we agree that the interpretation of DE and IE estimates after the first dose (Fig 1a and 1b) is compromised by the mixture of individuals who have received either one or two vaccine doses. Consequently, in the revised manuscript, we place little more weight on the estimates that report vaccine effectiveness after the second dose in a sample that includes only double-vaccinated individuals in the treatment group and report these estimates also in the abstract.

We summarize potential explanations for increasing DE and IE estimates in the revised manuscript and write that:

“We observe that the direct vaccine effectiveness estimates increase over time and stabilize around follow-up week eight. Thus, the stabilization of direct vaccine effectiveness estimates occurs later in our observational study than in several other observational studies.^{3,4,9,10} This tendency might be explained by several factors. First, and most importantly, our direct effectiveness estimates (Fig. 1a) include a mixture of individuals who have received either one or two vaccine doses. The additional protection conferred by the second vaccine dose is expected to be observed, at the earliest,

four weeks after the beginning of the follow-up as there was a minimum of three weeks interval between the first and second dose. Second, the later stabilization of direct vaccine effectiveness estimates might be explained by several contextual differences between the previously reported results and our findings including, among others, different testing strategies, study populations and differences in administrative health care register protocols.”

“The indirect effectiveness estimates increase gradually after the first vaccine dose, reflecting the increase in direct effectiveness estimates. However, as expected, the indirect effects among unvaccinated spouses are smaller than the direct effects among vaccinated individuals.”

2. Importantly, the authors make no mention of how these results might change in light of the emergence of the Delta variant. It is possible that the results could look very different if taking into account the recent period when the Delta variant has become dominant. While I understand that further extending the analysis would not be possible (i.e. the HCWs would all be vaccinated), the authors should explicitly discuss how the emergence of the Delta variant might affect both the DE and IE estimates in the Discussion.

We agree that it is important to discuss how our results might change in light of the Delta variant, as our results cover only a period before the Delta variant becoming the dominant strain. To address this remark, we have revised the Introduction and Discussion sections and write that:

“There is evidence that vaccinated individuals infected with SARS-CoV-2 might have lower viral loads than unvaccinated infected individuals, even though this might have changed after the emergence of the Delta variant.” (Introduction)

“After our study period, the SARS-CoV-2 Delta (B.1.617.2) variant has become the dominant strain in many parts of the world. While mRNA-based Covid-19 vaccines are still expected to lower the risk of infection and remain highly effective against severe disease, the emergence of the Delta variant has likely eroded the direct and indirect vaccine effectiveness by increasing the likelihood of breakthrough infections and viral transmission from vaccinated individuals who become infected.¹⁸ Further studies are required to understand whether booster vaccinations do not only improve direct vaccine effectiveness but also strengthen vaccine-associated reduction in transmission. Overall, there is a need for new evidence to understand how the indirect effects of Covid-19 vaccines on unvaccinated adults and children support the prospect of herd immunity and to inform questions related to vaccine booster strategies and the possible mass vaccination of children.” (Discussion)

3. Reviewer 1 asked for the number of infections broken down by subgroup, which could help to indicate the potential for confounding by the covariates included in the analysis. Including the total number of infections in the vaccinated vs unvaccinated groups is not really addressing the reviewer's suggestion.

Thank you for raising a question about the number of infections in different subgroups. After re-examining the comments by the Reviewer 1 and our responses, we agree that our previous response did inadvertently only partially address the reviewer's comments related to demographics and the number of infections by subgroup. To address this concern and the reviewer's suggestion we now report the number of infections by all subgroups (Tables S5, S6 and S7) and summarize this information in the main text (Results section: paragraph 4)

4. I am concerned about the causal language in the abstract where the authors write: "here we show that mRNA-based Covid-19 vaccines do not only protect vaccinated individuals.". It is possible the results could be due to confounding. Weaker language (e.g. "vaccination is associated with...") should be used.

We agree that the conclusion in the abstract made too strong claims about the potential causality of our results. However, the phrase "here we show" directly follows the journal's formatting guidelines and we have decided to keep this phrase. We now write in the abstract that:

"Here we show that mRNA-based Covid-19 vaccines are associated with a reduction in SARS-CoV-2 infections not only among vaccinated individuals but also among unvaccinated adult household members in a real-world setting."

5. As a sensitivity analysis, the authors assigned the start of the follow-up period with the start of the vaccination campaign. As pointed out in my previous review, this analysis has inherent bias due to the state of the epidemic over time. The authors reply that this analysis has been conducted in other studies, which is not a valid reason in my opinion (and I'm not aware of any such studies), and no references are provided. Also, they emphasize that the results from the sensitivity analysis are in agreement with the main analysis where a random follow-up date is used. I do not find this very reassuring, as it could represent evidence of potential confounding in the main analysis.

We agree that the sensitivity analysis using a follow-up period with the start of the vaccination campaign is problematic and have decided to delete the analysis altogether.

6. To explain the non-significant indirect effectiveness estimate for children, the authors say this may be due to “the notion that children could be less susceptible to SARS- CoV-2 from other household members than adults”. I don’t feel that this interpretation is necessarily correct. Why should this influence the RELATIVE risk for children living with a vaccinated vs unvaccinated HCW? I would think it should only affect the power to detect a significant effect, not the point estimate itself. Further explanation is needed.

We agree that associating the non-significant indirect effectiveness estimates for children with the notion that children could be less susceptible to SARS- CoV-2 from other household members than adults was too hasty. To address this concern, we have revised the sentence and now write that: “The power to detect statistically significant effects in children could be weakened by a smaller risk of SARS-CoV-2 infections among children than adults.”

7. The authors include a description of their data in the Introduction. While this is important information to include in the main text, I don’t think it's appropriate to include it in the Introduction. Either the manuscript should be reformatted to include the Methods before the Results section, or the essential methods (included in the last few paragraphs of the Introduction) should be incorporated into the Results section.

Thank you for pointing out that the description of our data is potentially inappropriately included in the Introduction. We agree that the description of the data and the description of essential methods should be incorporated into the Results section as there is no possibility for including a ‘Methods’ section before the ‘Results’ section in this journal (The formatting guide for *Nature Communications* says that authors should include ‘Methods’ section at the end of the text). To address the comment, we have reformatted the structure of our main text and now include a description of the data and essential methods in the ‘Result’ section. We have also explicitly informed the editor about these revisions to the structure of the main text and requested the editorial team to instruct us about the correct structure.

8. In the Discussion, the authors refer to “existing observational studies that have relied on comparisons of vaccinated individuals at different time periods before and after administration of a vaccine to their close contact” but do not provide any reference to these studies.

Thank you for pointing out missing references related to this statement. In the revised manuscript, we provide two references to support this statement. Based on our reading of articles by Shah et al. (2021) and Monge et al. (2021), these studies have relied adjusted comparisons of individuals at different time periods before and after administration of a vaccine to their close contact.

9. By randomly assigning a follow-up period for each unvaccinated HCW and their family members in the control group, I think the authors are inherently assuming that the vaccine coverage among HCWs increases approximately linearly during the study period, which is not necessarily the case here.

We agree that the random assignment of follow-up period for each unvaccinated HCW and their family members may lead to a situation that vaccine coverage increases more linearly in the control group. In fact, we have now examined the weekly number of vaccines administered to health care workers during our study period and observed that the weekly number of administered vaccines varied substantially from week to week depending on the supply of vaccines in Finland. Thus, we observe that the vaccine coverage does not increase linearly among the HWCs. However, it is worth emphasizing that our regressions control for time-variant factors (or common trends) using week fixed effects, and thereby our estimates account for the changes in the supply of vaccines and state of the epidemic over time.

We now discuss the discrepancy between the random assignment of follow-up period in the control group, the varying weekly rate of vaccine administration among the HWCs and the importance of including week fixed effects in our revised manuscript and write that:

“The random assignment of a follow-up period for each unvaccinated healthcare worker and their family member parallels an assumption that the vaccine coverage among increases linearly over time. In practice, the weekly number of vaccines administered to health care workers varied substantially during the study period depending on the supply of vaccines and timing of vaccine administration. Importantly, our regression models control for time-varying factors using week fixed effects, and thereby changes in the supply of vaccines and state of the epidemic over time.”

Minor comments:

Figure S1: The authors should clarify in the figure legend why the total cumulative number of infections decreases (i.e. it reflects the corresponding decrease in the number of individuals). Otherwise, the figure could be misleading.

Thank you for pointing this out. We have added a clarification related to this in the figure legend.

Figure S2: Y-axis should specify that this is a % of the population.

Thank you for pointing this out. Figure S2 now takes this into account.

Lines 155: the lower 95%CI is missing.

Thank you for noticing this. The missing CI is now added to the text.

Line 300: What do the authors mean by: “recorded in calendar week t and thereafter”?

Thank you for pointing this out. We have now formatted the sentence, as it should have been in the first place: “recorded in calendar week t”.

Line 302: Need to clarify that they do not control for characteristics of both the HCW and their contact.

Thank you for noticing this. The text related to regression model specification now explicitly mentions that the model does not control for the characteristics of the HCW and their contacts.

Figure 2: Infections are combined between weeks 2-5 since the numbers are low. How does this differ from the cumulative estimate for 5 weeks of follow-up? In other words, do the estimates for week 6 only include infections between weeks 5-6?

Thank you for pointing this out. The estimate for weeks 2-5 represents the average of cumulative vaccine effectiveness in weeks 2, 3, 4, and 5. This is now explicitly mentioned in the text.

Cumulative infections for week 6 include all infections, which have occurred before (and including) week 6. Therefore, the estimate for week 6 represents the cumulative vaccine effectiveness 6 weeks after the vaccination.

Lines 43-44: “Evidence that vaccinated individuals infected with SARS-CoV-2 have lower viral loads” might not be true for the circulating Delta variant.

This is an important comment. The text now takes the Delta variant into account and also cites relevant studies.

REVIEWERS' COMMENTS

Reviewer #2 (Remarks to the Author):

I thank the authors for their careful consideration of my comments. I'm satisfied with the revised manuscript.